# Adaptive Evolution of New Variants of Dengue Virus Serotype 1 Genotype V Circulating in the Brazilian Amazon

**DOI:** 10.3390/v13040689

**Published:** 2021-04-16

**Authors:** Geovani de Oliveira Ribeiro, Danielle Elise Gill, Edcelha Soares D’Athaide Ribeiro, Fred Julio Costa Monteiro, Vanessa S. Morais, Roberta Marcatti, Marlisson Octavio da S. Rego, Emerson Luiz Lima Araújo, Steven S. Witkin, Fabiola Villanova, Xutao Deng, Ester Cerdeira Sabino, Eric Delwart, Élcio Leal, Antonio Charlys da Costa

**Affiliations:** 1Laboratório de Diversidade Viral, Instituto de Ciências Biológicas, Universidade Federal do Pará, Belem 66075-000, Brazil; geovanibiotec@gmail.com (G.d.O.R.); fvillanova@gmail.com (F.V.); 2Departamento de Moléstias Infecciosas e Parasitárias and Instituto de Medicina Tropical, Faculdade de Medicina, Universidade de São Paulo, São Paulo 05403-000, Brazil; danielle_egill@yahoo.com (D.E.G.); va.morais@usp.br (V.S.M.); robertamarcatti@gmail.com (R.M.); sabinoec@gmail.com (E.C.S.); 3Public Health Laboratory of Amapa—LACEN/AP, Health Surveillance Superintendence of Amapa, Macapa 68905-230, Brazil; edcelhamanu@hotmail.com (E.S.D.R.); fredjulio@gmail.com (F.J.C.M.); farmarlisson@hotmail.com (M.O.d.S.R.); 4General Coordination of Public Health Laboratories of the Strategic Articulation, Department of the Health Surveillance, Secretariat of the Ministry of Health (CGLAB/DAEVS/SVS-MS), Brasília 70719-040, Brazil; emerson.araujo@saude.gov.br; 5Department of Obstetrics and Gynecology, Weill Cornell Medicine, New York, NY 10065-4850, USA; switkin@med.cornell.edu; 6Vitalant Research Institute, 270 Masonic Avenue, San Francisco, CA 94118-4417, USA; xdeng@vitalant.org (X.D.); EDelwart@Vitalant.org (E.D.); 7Department Laboratory Medicine, University of California San Francisco, San Francisco, CA 94143-0134, USA

**Keywords:** arbovirus, flavivirus, Dengue, phylogeography, evolution, Brazil, Amazon, genomic surveillance

## Abstract

Dengue virus (DENV) is a mosquito-borne viral pathogen that plagues many tropical-climate nations around the world, including Brazil. Molecular epidemiology is a growing and increasingly invaluable tool for understanding the dispersal, persistence, and diversity of this impactful virus. In this study, plasma samples (*n* = 824) from individuals with symptoms consistent with an arboviral febrile illness were analyzed to identity the molecular epidemiological dynamics of DENV circulating in the Brazilian state of Amapá. Twelve DENV type 1 (DENV-1) genomes were identified, which were phylogenetically related to the BR4 lineage of genotype V. Phylodynamics analysis suggested that DENV-1 BR-4 was introduced into Amapá around early 2010, possibly from other states in northern Brazil. We also found unique amino acids substitutions in the DENV-1 envelope and NS5 protein sequences in the Amapá isolates. Characterization of the DENV-1 BR-4 sequences highlights the potential of this new lineage to drive outbreaks of dengue in the Amazon region.

## 1. Introduction

Dengue and Dengue Hemorrhagic Fever are febrile diseases caused by four dengue virus types: DENV-1, DENV-2, DENV-3, and DENV-4, transmitted primarily by the *Aedes aegypti* mosquito. DENV is a single-strand positive-sense RNA virus belong to the Flaviviridae family. Its genome of about 11,000 nucleotides encodes a polyprotein which is cleaved into three structural proteins (capsid, membrane, and envelope) and seven non-structural proteins (NS1, NS2a, NS2b, NS3, NS4a, NS4, and NS5). Like other RNA viruses, the DENV RNA polymerase is error-prone during its replication process, favoring the accumulation of genetic changes quickly over time. This increase in genetic diversity is related to adaptive changes, such as avoidance of the host’s anti-viral immune response [1].

Dengue is one of the world’s most impactful tropical-climate diseases, infecting more than 390 million people annually in more than 100 countries and endemic areas [2]. Since its reintroduction in Brazil in 1982, DENV has grown to be a serious public health problem. Brazil now has the highest number of notified annual infections of DENV in the Americas [3,4,5]. The highest numbers of reported cases were observed between 2010 and 2015; cumulatively, during these years there were more than 5 million reported cases. Furthermore, the transmission of all four serotypes of DENV has been documented in Brazil [4,6].

DENV-1 is phylogenetically classified into five genotypes (I–V), with genotype V being the most prevalent genotype circulating in the Americas in recent years. In the late 1900s, there were two independent introductions of DENV-1 genotype V from India to the Americas. The two introductions occurred in the early 1970s and in the mid-1990s, respectively. The infections were initially in the Lesser Antilles of the Caribbean and then spread to South America, causing epidemics throughout Brazil and other countries [7,8]. The strains involved in the first, in the early 1970s, and the second, in the mid-1990s, waves spread south by way of Venezuela and Northern Brazil, respectively. Until the mid-1980s, the Caribbean was identified as the main epicenter of the spread of genotype V in America. In the following years, there was a continuous spread of DENV-1 genotype V in the Americas, resulting in the emergence of different local viral lineages [9].

The state of Amapá is located in the extreme north of Brazil, bordering Guyana. It has a typical tropical climate, few temperature variations (an average of 27 °C), an average annual relative humidity of 81% and an average rainfall of around 2600 mm. The driest period is between the months of September to November (quarterly rainfall below 200 mm) and the rainiest is between March and May (quarterly rainfall greater than 1000 mm) [10]. This climate is propitious to maintenance of the *Ae. aegypti* mosquito, the main vector of DENV [11].

The northern region has been a gateway for different viruses responsible for several epidemics in Brazil, such as Yellow fever virus [12,13], Chikungunya virus [14] and DENV [15,16]. Among the factors related to the emergence of arboviruses in this region are the following predominant variables: (1) Favorable climatic conditions for maintaining the *Ae. Aegypti* vector; (2) borders with other South American countries (Venezuela, Guyana, and Colombia) that also have a high incidence of arboviral diseases; (3) poor housing and living conditions; and (4) connectivity between urban and rural areas [17].

The goal of the present study was to update available information about the DENV genotypes present in the state of Amapá using molecular epidemiological techniques. Due to the connectivity between the potential circulation of arboviruses in the state of Amapá and vectors in the Amazon basin, this investigation will be of value to scientists and public health officials dealing with the control and prevention of infections by this arboviral pathogen.

## 2. Materials and Methods

### 2.1. DENV Samples

Blood plasma samples were collected from individuals who sought treatment for symptoms consistent with an arboviral febrile illness between 2013 and 2016 in the Brazilian state of Amapá. A total of 96, 240, 283, and 205 samples from 2013, 2014, 2015, and 2016, respectively, were accessible and chosen arbitrarily. Viral RNA was extracted using a MagNa Pure 2.0 Roche automatic nucleic acid extraction machine (MagNA Pure LC instrument, Roche Applied Science, Indianapolis, IN, USA). The reagent kits used for extraction were from the MagNa Pure LC Total Nucleic Acid Isolation Kit-High Performance, Version 8, by Roche (Roche Applied Science, Indianapolis, IN, USA). The protocol used was that specified in the kit instructions. Firstly, 200 μL of blood plasma were used from each sample for extraction. If the sample was less than 200 uL, PBS was added to the sample until a total volume of 200 μL was reached. The final elution volume for each sample was 60 μL. After extraction, the samples were stored in a −80 °C degree freezer. The samples were then submitted to a series of qPCR assays. First, the ZDC (Zika, Dengue, Chikungunya) Multiplex qPCR Assay by BIORAD (Bio-Rad Laboratories, Inc.; Hercules, CA, USA), was applied to all samples. The samples that showed negative results for the ZDC assay were then submitted to a pan-Flavivirus multiplex qPCR assay, using the primers and protocol described in the articles [18,19,20,21].

### 2.2. Complete Genomes Sequencing

The samples that showed positive results in the ZDC assay were submitted to NGS (Next Generation Sequencing). At first, the first-strand cDNA synthesis was performed with random primers and SuperScript III kit (Life Technologies, Grand Island, NY, USA); the second strand of cDNA was produced using the Klenow FRAGMENT kit (New England Biolabs, Ipswich, MA, USA). After, the reverse transcription product was used to Nextera XT library preparation (Illumina, San Diego, CA, USA). The paired-end, 300 pb sequences generated by MiSeq were demultiplexed using Illumina software. Assemblers were used for the reconstruction of viral genomes, including SOAPdenovo2, Abyss, meta-Velvet and CAP3, Mira and SPADES programs. The full or partial genomes were then evaluated using Geneious R8 (Biomatters, San Francisco, CA, USA).

### 2.3. Dataset Construction

Initially, all newly obtained genomes were submitted for genotyping using the online Genome Detective Virus Tool [22]. This revealed that all DENV genomes belonged to serotype 1, genotype V.

Next, 1255 sequences of envelope gene region were selected, and we choose only DENV-1 genotype V. This alignment, comprising 1485 nucleotides, was used to construct the phylogenetic analysis.

To the phylodymics analysis of DENV-1 in Brazil, a curated database was used. First we selected the envelope genomic region (1485 nucleotide length) and performed BLASTn search to retrieve more sequences closely related to Amapá sequences. The final alignment contained 88 sequences, including 16 sequences from other northern Brazil states.

### 2.4. Phylogenetic, Molecular Clock, and Phylogeographic Analysis

The sequences were aligned using the Mafft software package component, and subsequently visually inspected and manually adjusted. Maximum likelihood phylogenetic trees were estimated using PhyML (NNI algorithm) [23] and the substitution model was choose as suggested by jModelTest [24].

To study the spatiotemporal dynamics of DENV-1 introduction in Amapá state, Bayesian phylogeographic analysis was applied in the dataset of the envelope region. Prior to performing this analysis, TempEst software [25] was used to assess the temporal-signal of our data by estimating the root-to-tip regression from maximum likelihood trees previously constructed. Outlier sequences identified from the tip-to-root regression plots were removed from this dataset. Finally, we found a moderate signal (correlation coefficient = 0.7744), suggesting the dataset was appropriate for the estimation of temporal parameters.

Time-calibrate were then performed using Bayesian settings in Beast package software [26]. The calibration point was the date of isolation of each sample. For molecular clock and demographic model selection, we performed path-sampling (PS) and stepping stone (SS) analysis, by running 100 path steps of 1 million iterations each. The best fit was the relaxed lognormal clock in combination with the Bayesian skyline growth model. Other settings in Beauti were as follows: substitution model (GTR), base frequencies (Estimated), site heterogeneity model (Gamma + Invariant Sites), number of gamma categories [4], tree model (random starting tree), length of chain (100,000,000), echo state to screen every (10,000), log parameters every (10,000). Convergence and the effective sample size (ESS) > 200 were examined using Tracer v1.7.1 (http://beast.bio.ed.ac.uk/tracer, accessed on 10 February 2021). The analyses were performed in duplicate and were combined in LogCombiner v.1.8.4 software. A maximum clade credibility tree (MCC tree) was then summarized discarding the first 10% of the sample trees using Tree Annotator in the Beast package. FigTree software was used for visualization and editing of both Maximum likelihood and Bayesian trees.

### 2.5. Selection Pressure Analysis

Complete genome sequences were first screened for recombination events using GARD tool (genetic algorithm for recombination detection) [27]. Next, we tested for evidence of natural selection pressure using several methods in HyPhy [28], implemented by the Datamonkey 2.0 server [29]. We searched for evidence of positive selection using different methods. Pervasive selection (selection that occurs persistently along the whole phylogeny) was performed in three algorithms: SLAC (Single-Likelihood Ancestor Counting), FUBAR (Fast, Unconstrained Bayesian AppRoximation), and FEL (Fixed Effects Likelihood). Episodic adaptive selection (occurs heterogeneously across the tree branches) was evaluated using the MEME (Mixed Effects Model of Evolution) algorithm. In addition, we used the aBSREL (adaptive Branch-Site Random Effects Likelihood) algorithm to determine whether a specific lineage or clade(s) had been subject to selection.

## 3. Results

Of the 824 samples tested by ZDC-PCR assay, 788 were negative and 36 samples (4.3%) were positive (0 from 2013, 8 from 2014, 23 from 2015, and 5 from 2016). Of the 36 positive samples, 24 were DENV (5 from 2014; 16 from 2015; and 3 from 2016), 11 were CHIKV (3 from 2014; 7 from 2015; and 1 from 2016), and 1 was ZIKV (from 2016). Results of Pan-flavivirus qPCR assay showed that of the 788 samples that underwent this assay, 22 were positive (0 from 2013, 5 from 2014, 14 from 2015, and 3 from 2016). A total of 13 whole DENV genomes were obtained from these 24 positives after undergoing Next Generation Sequencing (NGS). Twelve of the 13 DENV genomes obtained were found to be DENV-1, genotype V from the years 2014 and 2015; the other DENV genome was found to be DENV-2. The 12 DENV-1 sequences were deposited in GenBank at NCBI access number MW377899-MW377910. A Maximum likelihood tree was constructed from the 12 newly obtained DENV-1 genotype V genomes collected in Amapá with other DENV-1 genomes obtained from GenBank and collected throughout Brazil and Latin America (*n* = 1255). The phylogenetic analysis based on envelope sequences showed that the DENV-1 Brazilian sequences grouped into four clades, named sequentially based on their date of introduction in Brazil: BR2 to BR5 (lineage BR1 contained a single sequence which was hidden in phylogeny) [30] (Figure 1). The Amapá sequences from this study clustered into lineage BR4, while other Amapá sequences obtained in the mid-2000s (access numbers MH450099 and MH450088) grouped into lineage BR3, suggesting a continuous introduction of DENV-1 strains in this state.

To expand our results, we used envelope-gene (*n* = 88), including more sequences, to estimate the regional phylodynamics of DENV-1 in northern Brazil. This envelope-based tree indicated that DENV-1 BR4 was introduced from Venezuela into Roraima and Amazonas states (northern Brazil) (probability = 1.0) (Figure 2). This same analysis suggested that the ancestor geographic location of Amapá sequences was from northern Brazil (probability = 0.94). In addition, one envelope sequence (MH401997) identified from the city of Oiapoque in 2015, a municipality in the state of Amapá on the border with Guyana, grouped in the same clade as our sequences, reinforcing that the BR4 lineage was prevalent in the state in this period and disseminated locally. Later, strains of the lineage BR4 spread to other regions of Brazil, such as the Northeast, Midwest, and Southeast, being detected until the present time [31].

We undertook a screen for sites under positive selection using different models implemented in the HyPhy framework. Table 1 shows the codons along the entire genomes under positive selection of lineage BR4 sequences detected at least by one in the three methods performed. Site under positive selection were detected in the E, NS1, NS3, and NS5 regions. In total, FEL, FUBAR, and MEME algorithms found seven sites under positive selection, and SLAC algorithm found four sites under positive selection. In addition to the above analyses of the BR4 lineage as a whole, the FEL approach can be used to compare different selection pressures within specific clades in the dataset. The Amapá clade was selected in the lineage BR4 phylogeny which was set as the background dataset. We found four different instances of positive/diversifying selection acting in Amapá sequences, in codons 892, 2907, 2950, and 2951. Although this finding is derived from one selection pressure method only, we found evidence of episodic diversifying selection on almost all branches of the Amapá clade using adaptive Branch Site REL (aBRSRel) methods.

Molecular characterization was also performed using coding region and we observed four amino acid differences in all Amapá sequences in comparison to other BR4 lineage sequences: S892R (NS1), D2907N (NS5), R2950S (NS5), E2951L (NS5), K2952R/K (NS5), K2953Q/P/K (NS5), and E2056R/E (NS5). It is interesting to note that South American DENV-1 had the canonical motif ^2950^REKKLGE^2956^ in the NS5 gene region which is located in the loop of RNA-dependent RNA polymerase (RdRpol). All sequences described here have a distinct motif in the RdRpol, except the sequence 612 (MW377909). Additionally, we found 6 synonymous substitutions exclusive of Amapá sequences, which all were transition type: 1827 (C->T), 1954(C->T), 2301(G->A), 6792(C->T), 9012(C->T), and 9027(A->G), occurring predominantly in third codon position.

## 4. Discussion

Our phylogenetic analysis supports that DENV-1 genotype V has been widely distributed in northern Brazil, and that other South American countries have contributed to its introduction into this country. The 12 new DENV-1 sequences were classified as genotype V, and grouped into lineage BR4 (previously called Clade Ib by Nogueira 2018 [32]). This result was expected, since this genotype has been the most prevalent in the Americas over the last 40 years [9]. Furthermore, our results summarize several aspects about BR4 lineage phylodynamics and its spread pattern in Brazil. Briefly, BR4 was introduced in Brazil through its northern region, coming from Venezuela around 2007 [32]. After its introduction, this lineage disseminated to several states in the northern region, including Amapá, in addition to other regions of Brazil, such as the Southeast, Midwest, and Northeast. Notably, the BR4 lineage remains persistent and has been identified in both southeast and northeast regions in recent years [31].

Our phylogenetic data corroborate the epidemiological and historical data of dengue cases available from Brazilian Government surveillance agencies and a study published by Salles et al. [33]. The first cases of DENV-1 in Amapá was registered during an outbreak between 1998 and 2001, and, since then, its endemicity was established in this state. The monitoring of circulating serotypes indicated that there has been a low and silent circulation of DENV-1 in the state of Amapá between 2002 and 2009, following a pattern observed in the North and throughout Brazil, with alternate transmission cycles influenced by the predominance of different serotypes [34]. A recent study showed that, after a period of low incidence (2002–2009), DENV-1 re-emerged in 2010 and was the most prevalent serotype in Brazil, including in the northern region [33]. The period with the highest incidence of DENV-1 in the northern region coincides with the tMRCA value of the Amapá clade, dated to 2010 (95% HPD = 2008–2010), suggesting that the BR4 lineage reached the Amapá state during the viral population expansion in the Brazil northern region in 2010. Furthermore, according to Salles et al., during 2011–2014 there was a decrease in the incidence of DENV-1 in the northern region, and from mid-2014 there was a further re-expansion of DENV-1 in this region [34]. From this, we hypothesized that, since its introduction (2010) until identification (2014/2015), this strain has been circulating in Amapá state, and the long-term permanence favors local adaptation of this lineage.

DENV, like other RNA viruses, exhibit a high mutation rate because its RNA-dependent RNA polymerase lacks poof reading activity. Consequently, DENV generates many viral variants able to adapt and escape from the host’s immune system. Indeed, our results show three sites positively-selected in DENV Amapá sequences. Two sites were in the envelope gene and one in the NS5 gene. Mutations in envelope sequences could potentially affect host cell-binding efficiency. In addition, we identified other substitutions, all them resulting in non-conservative amino acid substitutions. NS1 protein plays an essential role as a cofactor in viral replication [35,36,37], which shows the importance of determining whether the difference in amino acids between strain studied affects the antigenicity and biological activity of NS1 [3]. NS5 encodes the RdRp and it is the highly conserved in flaviviruses genomes. Because RdRpol is absent in mammalian cell, it has been target to design of new DENV inhibitors [38], so, mutation in the RdRpol could be related to escape of drug that target this essential viral enzyme. Further studies are required to evaluate the impact of these variation on viral fitness. Taken together, our findings suggest that DENV-1 has experienced many adaptive changes after its introduction into the local population. Historically, the Amazon has been a hotspot for the emergence of many dengue epidemics [6,16,39,40]. Therefore, it is noteworthy to identify lineages with potentially adaptive changes, as such changes can contribute to its continued circulation in the region and its potential to cause new dengue epidemics in regions where the virus is highly endemic.

In Amapá state, the sequences identified in the mid-2000s, during the first DENV-1 cases in the state, were related to the BR3 lineage of genotype V, unlike the sequences from our study, which are related to the BR4 lineage. This suggests that different introductions of DENV-1 occurred at various times in the last 20 years, causing a replacement of the predominant circulating strain in the state.

In conclusion, this study contributes to the understanding of DENV-1 circulation in Brazil, knowledge that aids in the development of more effective prevention and control efforts. An improved characterization of which serotypes of DENV are circulating in which regions not only helps with general DENV prevention and control efforts, but also identifies regions at risk for the severest forms of illness caused by these viruses.

## Figures and Tables

**Figure 1 viruses-13-00689-f001:**
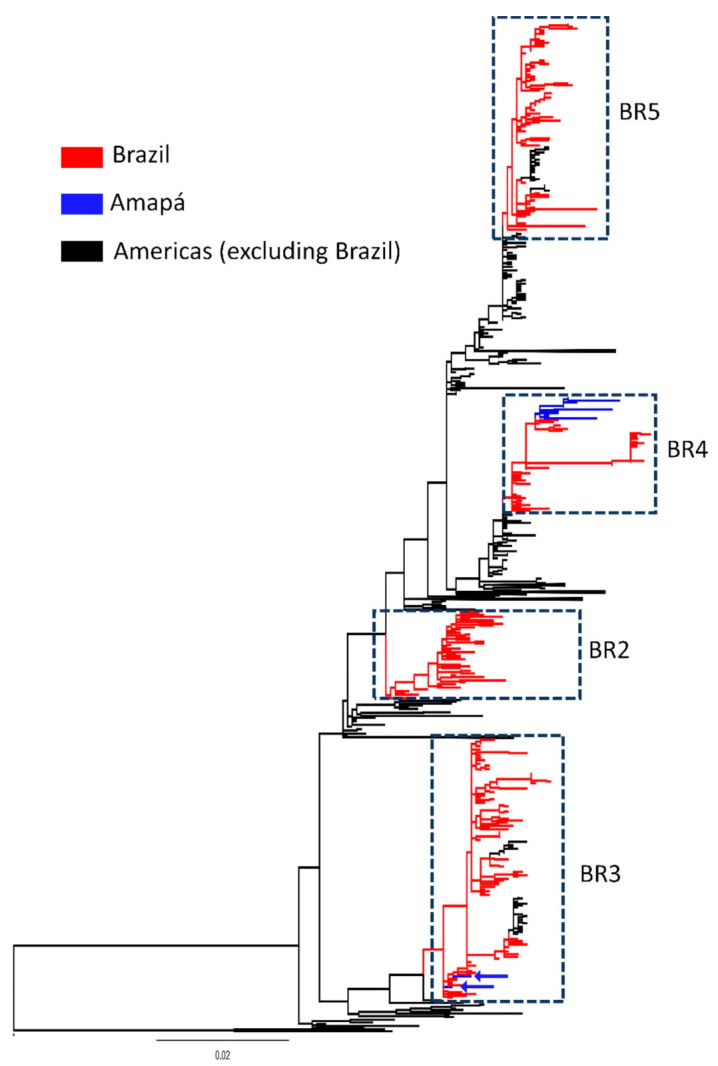
Phylogenetic tree from all DENV-1 genotype V envelope sequences deposited in GenBank database (*n* = 1255 sequences, l = 1485 nucleotides). The tree was constructed using the Maximum Likelihood approach and envelope sequences of DENV-1. Colors represent different sample locations. Amapá sequences are highlighted in blue, and the arrows signal the Amapá sequences obtained in mid-2000. The Brazilian lineages (BR2–BR5) are indicated by hatched rectangles. The tree was constructed using the software Phyml.

**Figure 2 viruses-13-00689-f002:**
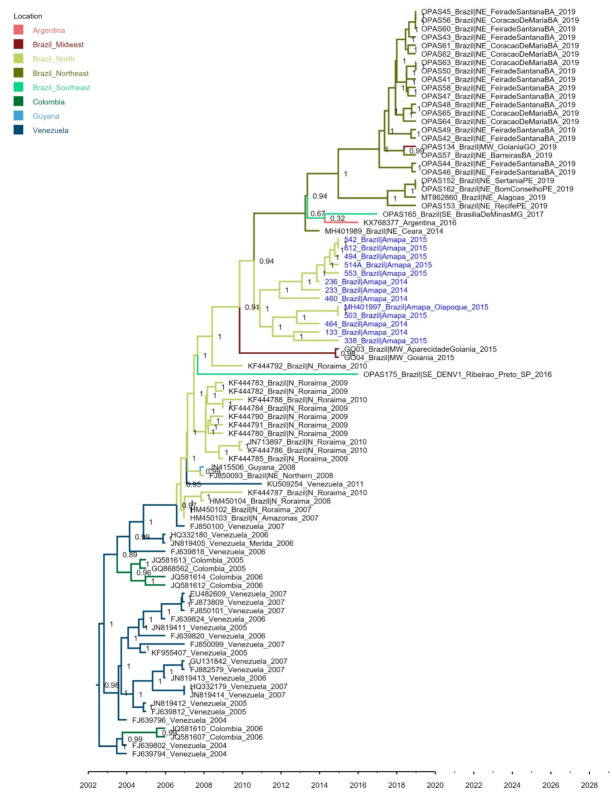
Time-scaled Bayesian phylogeny tree of the BR4 lineage of DENV-1. The tree was inferred using 88 sequences of DENV-1. The time-scale is shown. Amapá sequences are highlighted in blue. This tree was inferred using 88 sequences of DENV-1 genotype V. The branch colors refer to geographic location of collection of virus samples as well as inferred ancestral strains. Nodes numbers refer to its geographic probability. The maximum clade credibility (MCC) tree was inferred using Beast software.

**Table 1 viruses-13-00689-t001:** Codons under evidence for adaptive selection on the complete genome of DENV-1 lineage BR4.

Codon Position	Protein	FEL ^1^ (*p*-Value < 0.1)	SLAC ^2^ (*p*-Value < 0.1)	FUBAR ^3^ (Probability)	MEME ^4^ (*p*-Value < 0.1)
39	E	Yes	Yes	0.9793	Yes
193	NS1	Yes	Yes	0.9970	Yes
1684	NS3	Yes	Yes	0.9888	Yes
1752	NS3	Yes	Yes	0.9668	Yes
2951	NS5	Yes	No	0.9818	Yes
2913	NS5	Yes	No	0.9832	Yes
2997	NS5	Yes	No	0.9916	Yes

^1^ Fixed Effects Likelihood; ^2^ Single-Likelihood Ancestor Counting; ^3^ Fast, Unconstrained Bayesian AppRoximation; ^4^ Mixed Effects Model of Evolution.

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
