# Peer review of "Adaptive Evolution of New Variants of Dengue Virus Serotype 1 Genotype V Circulating in the Brazilian Amazon"

_viruses, 2021, doi:10.3390/v13040689_

Round 1

Reviewer 1 Report

Major comments:

-. The authors performed whole genome comparison as well as envelop sequence comparison. Please provide the reason of why two different comparative analyses are necessary to understand phylodynamics of DENV. Since phylodynamics is one of their focuses, please discuss topological congruency or discrepancy between Figure 1, Figure 2 and Figure 3 although individual datasets used for each tree are different from each other.

-. The authors claimed in the discussion section that their study is helpful in the development of more effective and prevention and control efforts. Please discuss how the results from this study would be helpful in detail in that efforts.

Minor comments:

Grammar/spelling check: Line 282, Ling 289.

Author Response

Major comments:

-The authors performed whole genome comparison as well as envelop sequence comparison. Please provide the reason of why two different comparative analyses are necessary to understand phylodynamics of DENV. Since phylodynamics is one of their focuses, please discuss topological congruency or discrepancy between Figure 1, Figure 2 and Figure 3 although individual datasets used for each tree are different from each other.

Resp: We deeply appreciate this comment and agree that showing distinct topologies seem quite confusing at first glance. The phylogenetic tree constructed using whole complete genomes (figure 2) has high branch support, but there are a very limited number DENV-1 complete genomes available from northern Brazil. So, we decided to use a lager dataset to fill this gap, although some branches have only moderate branch support (Figure 3). Overall, both trees harbor several topological congruencies between them, for example: 1) The Amapá sequences clade is closer to other northern sequences; 2) the sequence OPAS175_Ribeirao_Preto_2016 from southeast, did not group in any clade, and was positioned in branch sister of Northern sequences; 3) There are two clades from northeast sequences well defined; 4) the sequence OPAS165_BrasildeMinas from southeast is in the root of the northeast clades; 5) all Brazilian sequences are monophyletic in comparison to sequence from the other countries, as Venezuela and Colombia.

By other hand, the main discrepancy between these trees is the position of sequences from Midwest (GO03_AparecidadeGoiana_2015 and GO04_Goiania_2015), which in the complete genome tree they are positioned in near to northeast clade, with posteriori probability=1. While, in env tree they are in the root of Amapá sequences, with lower support in internal node (probability posteriori =0.94). Overall the incongruousness of our trees shows that to study the relatedness of sequences the inclusion of more variants is essecial. In the new version of the manuscript the tree constructed with genome was excluded because the estimated date range of these trees are the same.

-The authors claimed in the discussion section that their study is helpful in the development of more effective and prevention and control efforts. Please discuss how the results from this study would be helpful in detail in that efforts.

Resp: The characterization of new variants are important to provide information regarding its spread besides the possibility to design specific primers that can be used improve molecular surveillance.

The study of genetic variability of any pathogen has also a key role to improvement of the therapeutic strategies, vaccine design for example, and search of new drugs.

In addition, our finding has a direct impact in public health since we are showing the dissemination of a new variant in a region where Dengue is endemic.

Minor comments:

-Grammar/spelling check: Line 282, Line 289.

Resp: We have changed these sentences accordingly.

Reviewer 2 Report

The Authors described 12 new DENV-1 genotype V sequences—the most prevalent in the Americas over the last 40 years. The authors also confirmed the previously published epidemiological and historical data of dengue cases.

While data can have a local interest in Amapa and importance, the study does not have much novelty.

I would recommend publishing it as a shot report. Alternatively, the Authors should include detailed analyses of single nucleotide variations in new genomes (see below in Results).

Minor comments

Abstract

Line 29: “arboviral flavivirus” – this is a strange term.

Line 31-32: “viral pathogens” – Only DENV was mentioned above. Why “pathogens”?

Line 111: Which primers—random or virus-specific—were used for cDNA?

Line 113: please modify the sentence and “submitted”.

Line 122-124: It is difficult to understand this sentence. Please describe what was done step by step in separate sentences.

Materials and Methods

2.4. Phylogenetic Analysis:

The Authors only describe the first phylogenetic tree. Please describe how the second tree was constructed.

Major comments

Please use the uniform identification of serotypes (1, 2, 3, 4) and genotypes (I, II. II, IV, V) in the text. Now, there is no consistency in identification: DENV-1 (Line 34), DENV1 (Line 45), DENV=1 (Line 63), genotype 5 (Line 122).

Materials and Methods

2.2. Complete genomes sequencing:

Please provide more details on NGS data analyses, for example, how the consensus DENV genome sequences were identified and what was a threshold for single nucleotide (SNV) identification?

The general practice is to use two technical replicates for NGS; explain why it was not done.

Results

Please provide quantitative results on NGS depth and genome coverage.

A chapter for detailed analyses of SNVs in DENV genome sequences will be a great addition to the paper. While the Authors provide information on amino acid mutations, it has been recently shown that nonsynonymous mutations in flavivirus genomes also may affect infection phenotypes.

Please provide PCR and NGS data on Zika virus and Chikungunya virus (Line 107).

Table 1

Please provide a table legend and explain what is FEL, SLAC, FUBAR, and MEME.

Author Response

REVIEWER 2

-The Authors described 12 new DENV-1 genotype V sequences—the most prevalent in the Americas over the last 40 years. The authors also confirmed the previously published epidemiological and historical data of dengue cases.

While data can have a local interest in Amapa and importance, the study does not have much novelty.

-I would recommend publishing it as a shot report. Alternatively, the Authors should include detailed analyses of single nucleotide variations in new genomes (see below in Results).

Resp: We agree that this should be presented as a short communication but owing to the number of figures and tables we decided to apply as a full article

Minor comments

Abstract

-Line 29: “arboviral flavivirus” – this is a strange term.

Resp: We have modified this expression in the new version of the manuscript.

-Line 31-32: “viral pathogens” – Only DENV was mentioned above. Why “pathogens”?

Resp: We have corrected this phrase in the new version of the manuscript.

-Line 111: Which primers—random or virus-specific—were used for cDNA?

Resp: We used random primers applied to the metagenomics approach, this information was included in the new version of the manuscript.

-Line 113: please modify the sentence and “submitted”.

Resp: This sentence has been changed in the new manuscript.

-Line 122-124: It is difficult to understand this sentence. Please describe what was done step by step in separate sentences.

Resp: We are sorry for this confusion. We have shortened these procedures in the new version of the manuscript.

Materials and Methods

2.4. Phylogenetic Analysis:

-The Authors only describe the first phylogenetic tree. Please describe how the second tree was constructed.

Resp: The first tree is a maximum likelihood phylogeny constructed with the software PhyML. The second tree is an MCC phylogeny inferred with the Beast software and summarized using Logcombiner software.

Major comments

-Please use the uniform identification of serotypes (1, 2, 3, 4) and genotypes (I, II. II, IV, V) in the text. Now, there is no consistency in identification: DENV-1 (Line 34), DENV1 (Line 45), DENV=1 (Line 63), genotype 5 (Line 122).

Resp. We have changed this in the new manuscript.

Materials and Methods

2.2. Complete genomes sequencing:

-Please provide more details on NGS data analyses, for example, how the consensus DENV genome sequences were identified and what was a threshold for single nucleotide (SNV) identification?

Resp. We have used Ensemble Assembler previously describe (https://www.ncbi.nlm.nih.gov/pmc/articles/PMC4402509/) it takes into consideration the inherent diversity in viral genomes. All reads were first clustered into small contigs using a lenient alignment algorithm and were subsequently assembled into larger contigs using an iterative algorithm that merges contigs, filters low-quality bases, and corrects assembly errors. The resulting contigs for the assembly were ordered and oriented, corrected for frameshift errors, and annotated using in-house algorithms. Assemblies and annotations were then manually inspected, and any insertions/deletions (InDels) that were not resolved by frameshift correction algorithms and either were not supported by underlying reads or were in homopolymer regions were corrected. The full or partial genomes were then evaluated using Geneious R8 (Biomatters, San Francisco, CA, USA) for confirming results.

-The general practice is to use two technical replicates for NGS; explain why it was not done.

Resp. This wasn't an option for us because we have processed 824 samples using PCR. Although it seems to be an ideal practice, we are not sure if someone actually performs NGS in replicates.

Results

-Please provide quantitative results on NGS depth and genome coverage.

Resp.The methodology that we used and described in M&M includes only sequences with average genome coverage greater than 200x. For genome assembly, only readings with quality were considered Q scores of 30 (Q30). This approach is usually followed in most NGS protocols. The coverage of samples ranged from 211 to 1015.

-A chapter for detailed analyses of SNVs in DENV genome sequences will be a great addition to the paper. While the Authors provide information on amino acid mutations, it has been recently shown that nonsynonymous mutations in flavivirus genomes also may affect infection phenotypes.

Resp. We provide detailed information on amino acid substitutions because usually, it causes a great impact on the evolution of these new variants, mainly substitutions in conserved genes such as RdRpol. Synonymous mutations, on the other hand, have been little impact on the population dynamic of DENV, some nucleotides changes in the UTR of mRNA viruses have been described to increase viral fitness in cell culture but the significance of this to the spread of new lineages in host population is not clear.

-Please provide PCR and NGS data on Zika virus and Chikungunya virus (Line 107).

Resp. We have performed ZDC-PCR in 824 plasma samples and found 9 positives for chikungunya and 24 positives for Dengue. All ZDC-PCR positive samples were sequenced and in this manuscript, we describe the details of DENV. In the new version of the manuscript, we have included more information about these results. We have included this information in the new version of the manuscript.

Table 1

-Please provide a table legend and explain what is FEL, SLAC, FUBAR, and MEME.

Resp. We have included a description of these abbreviations in the table.

Round 2

Reviewer 1 Report

Thank you for improving the manuscript according to all the comments. I don't have more comments.

Reviewer 2 Report

The threshold for detection of mutations in NGS data is still not provided. If the Authors used a too low threshold, many identified mutations could be sequencing artifacts. Moreover, technical duplicates—the current experimentally validated standard for NGS data in virology—were not used that increases the chance of sequencing artifacts. Also, some samples had low coverage (e.g., 211); there are recent publications showing commin NGS artifacts in samples with coverage less than 400.

Detailed analysis of individual SNVs is not provided.

I recommend rejecting the pare as an Article. Or ask the Authors to modify it as Short Communication.